# Potential of *Trichoderma virens* HZA14 in Controlling *Verticillium* Wilt Disease of Eggplant and Analysis of Its Genes Responsible for Microsclerotial Degradation

**DOI:** 10.3390/plants12213761

**Published:** 2023-11-03

**Authors:** Ali Athafah Tomah, Iman Sabah Abd Alamer, Arif Ali Khattak, Temoor Ahmed, Ashraf Atef Hatamleh, Munirah Abdullah Al-Dosary, Hayssam M. Ali, Daoze Wang, Jingze Zhang, Lihui Xu, Bin Li

**Affiliations:** 1State Key Laboratory of Rice Biology and Breeding, Key Laboratory of Molecular Biology of Crop Pathogens and Insects, Institute of Biotechnology, Zhejiang University, Hangzhou 310058, China; ali_athafah@uomisan.edu.iq (A.A.T.); emansabah29@yahoo.com (I.S.A.A.); arifalikh2020@hotmail.com (A.A.K.); temoorahmed@zju.edu.cn (T.A.); jzzhang@zju.edu.cn (J.Z.); 2Plant Protection, College of Agriculture, University of Misan, Al-Amarah 62001, Maysan Province, Iraq; 3Plant Protection, Agriculture Directorate, Al-Amarah 62001, Maysan Province, Iraq; 4Xianghu Laboratory, Hangzhou 311231, China; 5Department of Botany and Microbiology, College of Science, King Saud University, P.O. Box 2455, Riyadh 11451, Saudi Arabia; ahatamleh@ksu.edu.sa (A.A.H.); almonerah@ksu.edu.sa (M.A.A.-D.); hayhassan@ksu.edu.sa (H.M.A.); 6Hangzhou Rural Revitalization Service Center, Hangzhou 310058, China; wdz2005@163.com; 7Institute of Eco-Environmental Protection, Shanghai Academy of Agricultural Sciences, Shanghai 201403, China

**Keywords:** *T. virens*, mycoparasitism, microsclerotial degradation, transcriptome, qPCR, *Solanum melongena* L.

## Abstract

*Verticillium dahliae* is a soilborne fungal pathogen that causes vascular wilt diseases in a wide range of economically important crops, including eggplant. *Trichoderma* spp. are effective biological control agents that suppress a wide range of plant pathogens through a variety of mechanisms, including mycoparasitism. However, the molecular mechanisms of mycoparasitism of *Trichoderma* spp. in the degradation of microsclerotia of *V. dahliae* are not yet fully understood. In this study, the ability of 15 isolates of *Trichoderma* to degrade microsclerotia of *V. dahliae* was evaluated using a dual culture method. After 15 days, isolate HZA14 showed the greatest potential for microsclerotial degradation. The culture filtrate of isolate HZA14 also significantly inhibited the mycelial growth and conidia germination of *V. dahliae* at different dilutions. Moreover, this study showed that *T. virens* produced siderophores and indole-3-acetic acid (IAA). In disease control tests, *T. virens* HZA14 reduced disease severity in eggplant seedlings by up to 2.77%, resulting in a control efficacy of 96.59% at 30 days after inoculation. Additionally, inoculation with an HZA14 isolate increased stem and root length and fresh and dry weight, demonstrating plant growth promotion efficacy. To further investigate the mycoparasitism mechanism of *T. virens* HZA14, transcriptomics sequencing and real-time fluorescence quantitative PCR (RT-qPCR) were used to identify the differentially expressed genes (DEGs) of *T. virens* HZA14 at 3, 6, 9, 12, and 15 days of the interaction with microsclerotia of *V. dahliae*. In contrast to the control group, the mycoparasitic process of *T. virens* HZA14 exhibited differential gene expression, with 1197, 1758, 1936, and 1914 genes being up-regulated and 1191, 1963, 2050, and 2114 genes being down-regulated, respectively. Among these genes, enzymes associated with the degradation of microsclerotia, such as endochitinase A1, endochitinase 3, endo-1,3-beta-glucanase, alpha-N-acetylglucosaminidase, laccase-1, and peroxidase were predicted based on bioinformatics analysis. The RT-qPCR results confirmed the RNA-sequencing data, showing that the expression trend of the genes was consistent. These results provide important information for understanding molecular mechanisms of microsclerotial degradation and integrated management of *Verticillium* wilt in eggplant and other crops.

## 1. Introduction

*Verticillium dahlia*, a soilborne ascomycete fungus, infects over 400 vascular plants and causes *Verticillium* wilt, a destructive disease of food and industrial crops [1,2]. Eggplant (*Solanum melongena* L.) is particularly susceptible to *Verticillium* wilt, which can cause significant yield and quality losses. Controlling *V. dahliae* is challenging due to its formation of resilient, dormant microsclerotia that can survive in soil for over a decade without a host plant [3]. Meanwhile, microsclerotia are composed of thick-walled dormant mycelia, which originate from swollen, septate hyphae by a process of budding and are resistant to some extreme environmental conditions, such as UV irradiation, cold temperatures, enzymatic lysis, and desiccation [4,5,6]. Additionally, completely resistant varieties are not yet available in production, and crop rotation is difficult due to limited cultivation options. Methyl bromide can reduce *Verticillium* inoculum in soil; however, its use is restricted due to its harmful impacts on the environment and human health [7,8].

Biological control is a promising approach for controlling *Verticillium* wilt of eggplant because biocontrol agents are environmentally friendly, highly specific to the target pathogen, and less likely to lead to resistance development. *Trichoderma* spp. are promising biocontrol agents for *Verticillium* wilt because they possess a variety of antagonistic mechanisms, including mycoparasitism, competition for space and nutrients, and the release of active secondary metabolites [9]. They are also able to produce cell wall-degrading enzymes (CWDEs), such as proteases, glucanases, and chitinases, which can kill or inhibit the growth of other pathogenic fungi [10,11]. In particular, high levels of hydrolytic enzymes produced by *Trichoderma* spp. may induce the expression of related genes, resulting in the microsclerotial degradation of *V. dahliae*.

Mycoparasitism is a type of parasitism in which one fungus parasitizes another, especially a plant pathogen. Recently, many biocontrol agents have been studied in various species of *Hypocrea* [12]. For example, Harman et al. [9] have elucidated the mycoparasitism mechanism of most *Trichoderma* strains. *Trichoderma* mycoparasitism occurs in three steps: (i) Trichoderma hyphae grow towards the target fungus and bind to its cell wall using lectins, (ii) Trichoderma hyphae attach to and coil around the target hyphae, forming appressoria, and (iii) Trichoderma appressoria penetrate the target hyphae and release fungi-toxic enzymes, such as chitinases, glucanases, and proteases, to kill or degrade the target fungus [13,14,15].

The activity of these enzymes causes the degradation of the cell walls of the other fungus, which finally results in the killing of the target host. In particular, high levels of hydrolytic enzymes produced by *Trichoderma* spp. may cause microsclerotial degradation of *V. dahliae*. Genome-wide expression studies are a promising new approach for identifying genes associated with mycoparasitism [16]. Transcriptome analysis using expressed sequence tags (ESTs) has identified many active genes involved in the response of Trichoderma strains to diverse stress conditions, as well as the role of mycoparasitism-related genes in the confrontation between different Trichoderma strains and host fungi [17,18]. Gene analysis revealed that several genes are involved in the expression of CWDEs: the production of alkaline proteinase is associated with gene *Prb1* in *T. harzianum* [19], the production of β-1,4-Endoglucanase is associated with gene *Egl1* in *T. longibrachiatum* [20], the production of chitinase is associated with gene *Chit33* in *T. harzianum* [21], the production of β -1,3-glucanase enzyme is associated with the gene *Chit33* in *T. virens*, and the production of β-1,6-glucanase enzyme is associated with the gene *TvBgn3* in *T. virens* [17].

The aim of this study was to screen *Trichoderma* isolates with the ability to control *Verticillium* wilt disease of eggplant and to identify genes encoding enzymes involved in microsclerotia degradation based on the analysis of both transcriptomics and quantitative reverse transcription PCR (qRT-PCR).

## 2. Results

### 2.1. Dual Culture Assay

The antagonistic activities of the 15 *Trichoderma* spp. isolates toward *V. dahliae* were determined using a dual plate assay on PDA media. All tested isolates exhibited varying degrees of antagonistic activity against *V. dahliae* (Figure 1). Isolate HZA14, which belongs to class 1, had the highest antagonistic activity. On the PDA plate, isolate HZA14 completely overgrew the colonies of the pathogen (Figure 1N), causing the microsclerotia to disappear. By comparison, isolates of HZA1, HZA2, HZA10, HZA11, HZA12, and HZA13 with higher antagonistic activity belong to class 2 (Figure 1). The isolates of HZA3, HZA4, HZA5, HZA6, HZA7, HZA8, and HZA15 with slightly antagonistic activity belong to class 3 (Figure 1).

### 2.2. Effect of Culture Filtrates of Trichoderma Isolates on Mycelial Growth and Conidial Germination of V. dahliae

The activity of culture filtrates produced by *Trichoderma* isolates against the mycelial growth of *V. dahliae* was assessed by preparing plates containing culture filtrates of different concentrations. Among the 15 isolates, only isolate HZA14 at different dilutions showed significant inhibition of the mycelial growth of *V. dahliae* (Figure 2A), and the inhibitory effect increased with the increase of the culture filtrate concentration. Indeed, the mycelial growth was completely inhibited at a culture filtrate concentration of 100%. To evaluate the inhibitory activity of culture filtrates at different concentrations against the germination of conidia, a well diffusion assay was performed. Results from this study indicated that there was a significant difference (*p* < 0.05) among inhibition zone diameters (Table 1), while the inhibitory effect increased with the culture filtrate concentration, and complete inhibition was achieved at a concentration of 100% (Figure 2B).

### 2.3. Assessment of Siderophore and IAA Production

The CAS assay was used to determine siderophore production. Only the tested *T. virens* HZA14 isolate showed a positive result, indicating that it is capable of producing siderophores. Orange halos were visible around the hyphae of isolate HZA14 grown on CAS media for 3 days (Figure 3), showing the ability of *T. virens* HZA14 to produce siderophores. For assessing the ability to produce IAA, the *T. virens* HZA14 was cultured in PD broth with tryptophan for 15 days. The pink color change was found after the mixture of culture supernatants with the Salkowski reagent, indicating that the *T. virens* HZA14 had the ability to produce IAA (Figure 3C). To quantify the IAA, IAA concentrations were determined. The maximum concentration was 218.61 ± 1.96 μg/mL after 15 days of incubation, followed by 138.48 ± 1.45 μg/mL for 10 days of incubation and 112.87 ± 1.74 μg/mL for 5 days of incubation (Figure 3D). These results indicate that IAA content produced by *T. virens* HZA14 increased with the increase of the incubation time.

### 2.4. Control Efficiency of T. virens HZA14 against Verticillium Wilt

The potential of *T. virens* HZA14 in controlling *Verticillium* wilt disease was assessed in pot experiments, with inoculation conducted at different time intervals. The results of controlling assays showed that inoculation with *T. virens* HZA14 reduced the incidence and disease severity of eggplant *Verticillium* wilt compared to the control (Figure 4). According to the results of statistical analysis, the effect of 30 days after inoculation with *T. virens* HZA14 was better than that of 10 and 20 days after inoculation with *T. virens* HZA14. Indeed, the maximum reduction of disease severity (02.77 ± 0.62%) was found at 30 days after sowing, followed by 23.88 ± 0.62% at 20 days after sowing and 44.43 ± 0.95% at 10 days after sowing, while the control efficacies were 96.59 ± 0.76%, 67.99 ± 0.83%, and 39.10 ± 1.30%, respectively (Table 2).

### 2.5. Plant Growth Promotion (PGP)

In the greenhouse pot experiment, 30 days after sowing, eggplant seedlings treated with *T. virens* isolate HZA14 exhibited a significant increase in stem and root length, as well as overall plant height, compared to untreated control plants (Figure 5A–D).

In addition, the statistical analysis showed that inoculation with *T. virens* isolates HZA14 significantly increased the lengths of stem and root by 16.54 and 13.50%, respectively (Table 3). Meantime, inoculation with *T. virens* isolate HZA14 significantly increased the fresh weight (biomass) of stem and root by 17.20% and 28.00%, respectively. Furthermore, inoculation with *T. virens* isolate HZA14 significantly increased the dry weight of stem and root by 20.31% and 54.55%, respectively (Table 3).

### 2.6. Microsclerotial Degradation of V. dahliae

To investigate the additional potential of isolating HZA14 in countering microsclerotial degradation by *V. dahliae*, a dual culture technique was employed. After three days of incubation, the hyphae of isolate HZA14 had completely enveloped the *V. dahliae* colony. However, the black microsclerotia produced by *V. dahliae* persisted in a circular zone and exhibited limited degradation (Figure 6A). Six days after incubation, microsclerotia began to degrade. Subsequently, the number of microsclerotia decreased significantly from 9 to 12 days of incubation. However, 15 days after incubation, microsclerotia were hardly observed, showing that *T. virens* isolate HZA14 has the potential to degrade microsclerotia of *V. dahliae*. Similarly, the process of microsclerotial degradation could be seen by reversing the petri dish (Figure 6B).

### 2.7. Transcriptome Data by Using RNA Sequencing

A total of fifteen libraries yielded sequences across five distinct periods. High-quality, clean reads were generated through a rigorous filtering process, which excluded low-quality reads (1%), reads containing Ns (0%), and those contaminated by adapters (2%) (Appendix A). The analysis generated more than 695 million raw reads, with an average of 46.3 million reads in each sequencing library (Appendix A). The total of the clean reads was 672.2 million, with an average of 44.8 and 97% raw clean reads in each sample. The total of the mapped reads was more than 541 million, with an average of more than 36 million, and there were 79.91% clean mapped reads in each sample.

### 2.8. Correlation Analysis of RNA-Seq Data

The clean reads were spliced and aligned to the reference *Trichoderma virens* Gv29-8 genome retrieved from the *Trichoderma* genome website (https://www.ncbi.nlm.nih.gov/nuccore/ABDF02000077.1 accessed on 5 July 2022) using TopHat and assembled by Cufflinks software ver. 2.2.1. The effects of sequencing depth and gene length on the mapped read counts were estimated by calculating the fragments per kilobase per transcript per million mapped reads (FPKM) values. To illustrate the gene expression profile among samples, genes expressed in 15 samples were assayed by cluster analysis, and a heatmap was drawn (Figure 7). Gene expression exhibited variations across the five distinct time periods. The consistency of the three replicates within each treatment group partially substantiated the reliability of the RNA-seq data.

### 2.9. DEGs Analysis

Based on analysis of the differentially expressed genes (DEGs) (Figure 8), the biggest number of DEGs (1914 up-regulated genes and 2114 down-regulated genes) occurred after 15 days (15 d vs. 3 d), while the minimal number of DEGs (1197 up-regulated genes and 1191 down-regulated genes) occurred after 6 days (6 d vs. 3 d). After 9 days, the DEGs include 1758 up-regulated genes and 1963 down-regulated genes (9 d vs. 3 d), while after 12 days, the DEGs include 1936 up-regulated genes and 2050 down-regulated genes (12 d vs. 3 d). The detailed information of the DEGs was able to be visualized by a volcano plot (Appendix A).

### 2.10. Gene Ontology Classification of Differentially Expressed Genes

Gene Ontology (GO) functional enrichment analysis was employed to identify and characterize the primary biological functions and pathways in which the DEGs participate. The enrichment analysis for DEGs revealed that the down-regulated genes were more than the up-regulated genes in cellular components, biological processes, and molecular functions (Figure 9). In cellular components (Appendix A), the mainly enriched terms included membrane, organelle, organelle part, and cell part (6 d vs. 3 d); protein-containing complex, organelle, organelle part, membrane part, and cell part (9 d vs. 3 d); membrane, protein-containing complex, organelle, organelle part, membrane part, cell part (12 d vs. 3 d); and membrane, protein-containing complex, organelle, organelle part, membrane part, cell part (15 d vs. 3 d). In the biological process, the mainly enriched terms included biological regulation, cellular process, localization, metabolic process, and response to stimulus (6 d vs. 3 d); biological regulation, cellular process, localization, metabolic process and response to stimulus (9 d vs. 3 d); biogenesis, metabolic process, biological regulation, cellular component organization, cellular process, localization, and response to stimulus (12 d vs. 3 d); and biogenesis, biological regulation, cellular component organization cellular process, localization, metabolic process, and response to stimulus (15 d vs. 3 d). In molecular function, the mainly enriched terms included catalytic activity, transporter activity, and binding (6 d vs. 3 d); catalytic activity, binding, and transporter activity (9 d vs. 3 d); binding, catalytic activity; molecular function regulator, transporter activity, and transcription regulator activity (12 d vs. 3 d) and binding, catalytic activity; molecular function regulator, transporter activity, and transcription regulator activity (15 d vs. 3 d).

### 2.11. KEGG Enrichment Analysis of DEGs

Kyoto Encyclopedia of Genes and Genomes (KEGG) pathway annotation and enrichment analysis were used to evaluate significantly enriched DEGs using the false discovery rate (FDR ≤ 0.05). During the interaction between *T. virens* and *V. dahliae*, the KEGG pathway analysis revealed that the DEGs were distributed in different metabolic pathways (Figure 10). In four interaction stages, the metabolic pathways of ribosomes all contained the highest DEGs, the carbon metabolite pathways also all had relatively high DEGs, and the other pathways appeared in different stages. In 6 d vs. 3 d, the metabolite pathways were associated with the glyoxylate and dicarboxylate metabolism and indole alkaloid biosynthesis.

In 9 d vs. 3 d, more metabolite pathways were involved, including glyoxylate and dicarboxylate metabolism, fructose and mannose metabolism, an indole alkaloid biosynthesis, pyruvate metabolism and glutathione metabolism. In 12 d vs. 3 d, just glyoxylate and dicarboxylate metabolism pathways were found. However, in 15 d vs. 3 d, metabolism pathways only had glyoxylate and dicarboxylate metabolism, fructose and mannose metabolism, and glutathione metabolism (Figure 10). These results indicate that the microsclerotia degradation of *V. dahliae* might be involved in the enzymes of different pathways.

### 2.12. Analysis of DEGs Related to Enzymes of Microsclerotial Degradation

The cell wall is a distinctive component of fungi, primarily comprised of glucans, chitins, and glycoproteins, while microsclerotia is composed of dense mycelia. Consequently, genes encoding cell wall-degrading enzymes (CWDEs) are associated with the enzymatic processes involved in microsclerotial degradation. In addition, alpha-N-acetylglucosaminidase, laccase-1, and peroxidase have also been involved in microsclerotial degradation. Therefore, the microsclerotial degradation may be associated with the DEGs encoding endochitinase A1, endochitinase 3, endo-1,3-beta-glucanas, alpha-N-acetylglucosaminidase, laccase-1, and peroxidase. The statistical analysis indicated that six enzyme-coding genes were all up-regulated, but the Log2FC values of enzyme-coding DEGs varied with incubation time in the process of interaction between the hyphae of *T. virens* HZA14 and the microsclerotia of *V. dahliae* (Figure 11). As examples, the Log2FC values of endochitinase A1-coding DEG were 3.39, 7.42, 11.81, 18.27, and 3.51; the Log2FC values of endochitinase 3 were 2.54, 2.51, 14.82, 7.87, and 8.37; the Log2FC values of endo-1,3-beta-glucanase were 15.11, 21.42, 42.02, 12.42, and 12.40; the Log2FC values of alpha-N-acetylglucosaminidase were 4.83, 12.52, 11.64, 11.56, and 4.04; the Log2FC values of laccase-1 were 34.87, 33.19, 64.58, 65.09, and 22.98; and the Log2FC values of peroxidase were 56.37, 53.32, 67.94, 85.51, and 55.59, respectively, after 3, 6, 9, 12, and 15 days of incubation.

### 2.13. Detection of RT-qPCR

RT-qPCR analysis was used to validate the DEGs in the mycoparasitic process between *T. virens* HZA14 and the microsclerotia of *V. dahliae* for different interaction stages in PDA medium. The expression of six enzyme-coding genes, including endochitinase A1 (XM_014096114.1), endochitinase 3 (XM_014099969.1), endo-1,3-beta-glucanase (XM_014101914.1), alpha-N-acetylglucosaminidase (XM_014095377.1), laccase-1 (XM_014095593.1), and peroxidase (XM_014105606.1), possibly involved in microsclerotial degradation, was confirmed by RT-qPCR, and the actin gene was used as an internal reference gene to normalize gene expression (Figure 12). The results showed that these gene expression profiles were similar to those of the DEGs in transcriptome analysis, confirming the reliability of transcriptome data (Figure 11).

## 3. Discussion

*Verticillium* wilt disease, caused by *V. dahliae*, is a major disease that severely affects eggplant production in China [22]. The *Trichoderma* species has been widely used as a biological control agent against many important plant fungal pathogens [23]. The successful application of *Trichoderma* species in suppressing plant pathogens depends on many mechanisms that work directly or indirectly [24]. In the current study, a total of 15 *Trichoderma* isolates were tested, and *T. virens* HZA14 exhibited an antagonism to *V. dahliae* in class I, which was consistent with the results reported by D’ercole et al. and Reghmit et al. [25,26]. The stereomicroscope showed that the microsclerotia of *V. dahliae* were degraded during their interaction, which is consistent with the results studied by Jabnoun-Khiareddine et al. [27]. Moreover, in agreement with the data of [28,29,30], the results indicated that metabolites of *T. virens* HZA14 were involved in the biocontrol activity against *V. dahliae* by inhibiting mycelial growth and conidia germination. As we know, mycoparasitic is an important action mode of *Trichoderma* spp. against microsclerotial degradation, mycelial growth, and conidia germination of *V. dahliae* [31]. *Trichoderma* species have demonstrated effectiveness in the management of soil-borne fungal diseases. Their efficacy can be attributed to a diverse array of mechanisms, including mycoparasitism, antagonism, and competitive interactions, which can be harnessed to combat pathogenic fungi [32].

In this study, the severity of *Verticillium* wilt disease in eggplant seedlings was significantly reduced when the pot soil was inoculated with conidia suspensions of *T. virens* at various time points, in contrast to the negative control. Our findings align with previous studies that have demonstrated the antagonistic activities of different *Trichoderma* species against *V. dahliae* isolated from diverse plant sources. In the study by D’ercole et al. [25], *T. viride* isolates T46 and T117 caused a 30% reduction in disease incidence of *Verticillium* wilt in eggplant. *T. harzianum* significantly reduced the colonization of *V. dahliae* on potato stems, thereby increasing total potato yield by 15.7% [33]. *T. harzianum*, *T. viride*, and *T. virens* isolates reduced the disease index of *Verticillium* wilt by more than 80% in tomatoes in Tunisia [27]. *T. longibrachiatum* T7 successfully controlled *Verticillium* wilt disease on tomatoes with a 90.71% reduction of disease severity in Algeria [34]. In olive, *T. asperellum* strains Bt3 and T25 significantly reduced disease severity caused by highly virulent *V. dahliae* in Spain and other Mediterranean countries [35]. Nowadays, over 35 *Trichoderma*-based biofungicides against soilborne and foliar diseases are registered [36]. The product Remedier WP (containing mycelium and spores of *T. harzianum* and *T. viride*) reduced *Verticillium* wilt in strawberries in Poland [37]. Yao et al. [38] reported that the available *Trichoderma* products in the current market include Trichodex, a commercial preparation of *T. harzianum* T-39 (Bioworks, MA, USA); RootShield, a commercial preparation of *T. harzianum* T-22 (Bioworks, MA, USA); Binab TF, a mixed-agent of *T. harzianum* and *T. polyspora* (Binab Bio Innovation AB, Helsingborg, Sweden); and Sentinel, a commercial preparation of *T. atrovilide* (Novozymes, Hallas Alle, Kalundborg, Denmark).

The findings of this study revealed that *T. virens* HsZA14 exhibited the capability to produce siderophores and 3-indoleacetic acid. (Figure 3A,C). Many of the *Trichoderma* strains have shown the ability to produce siderophores on a CAS agar medium and indole-3-acetic acid in PD broth supplemented with tryptophan [39,40]. Siderophores convert Fe^3+^ ions to Fe^2+^ ions, which plants can use and then efficiently transport from the roots to the shoots. Indole-3-acetic acid has been reported to be able to stimulate plant growth and development [41]. The production of the plant growth hormone 3-indoleacetic acid (IAA) has been reported to play an important role in enhancing plant growth in many *Trichoderma* strains [42]. In addition, *T. asperellum* Q1 was found to be able to promote the growth of *Arabidopsis thaliana* in an iron-deficient or insoluble iron-containing (Fe_2_O_3_) medium [43]. Numerous studies have reported that some *Trichoderma* strains could improve plant growth and development through the production of growth stimuli such as siderophores and IAA [44]. The treatment with *T. harzianum* and *T. asperellum* promoted the plant growth parameters, such as the fresh and dry weight of shoot and root and plant height in chili [45]. In this present study, the growth promotion in eggplant seedlings was clearly identified.

Microsclerotia in *V. dahliae* originate through budding from septate, swollen hyphae comprised of dense, thick-walled, melanized cells. These structures have the remarkable ability to endure in a dormant state, persisting from 10 days to as long as 30 years within agricultural soils [46,47,48]. *V. dahliae* has evolved complex mechanisms to achieve pathogenicity in plants, mainly including the germination and growth of microsclerotia, infection, and successful colonization [49]. Consequently, the degradation process of microsclerotia is a critically important way of breaking the disease cycle. Mycoparasitism is thought to be the most important direct way that *Trichoderma* species can reduce the number of pathogens [50,51]. Mycoparasitism is an ancestral trait in *Trichoderma* strain that involves chemotrophic growth, recognition, and coiling, as well as the interaction of hyphae with the secretion of specific lytic enzymes [51]. Chitin and glucan are the major constituents of cell walls in many fungal pathogens [52]. Correspondingly, *Trichoderma* strains are usually defined by their ability to secrete lytic enzymes like chitinases, glucanases, and proteases that can break down the cell walls of pathogenic fungi during the mycoparasitic process [53].

Genome-wide expression analysis was demonstrated for many *Trichoderma* species toward plant-fungal pathogens [18,54,55]. For example, Illumina sequencing is used to find differentially expressed genes, which makes it possible to identify the expression profiling of biosynthesis-related genes [56]. In light of that, one study has been carried out to look at how *T. atroviride* attacks the hyphae of *Verticillium* using the microarray method, but it did not focus on the microsclerotia [57]. However, the molecular mechanism of biocontrol agent *T. virens* during the mycoparasitic action on the microsclerotia of *V. dahliae* is still unclear.

Our study provides transcriptome information for the gene expression of *T. virens* HZA14 during mycoparasitism of *T. virens* on microsclerotia in different periods of time. Based on the analysis of the transcriptome, 12,037 genes were expressed differently during *T. virens* mycoparasitism at five different times. The number of DEGs was very close to that of annotated genes (12,428) in whole genomes of *T. virens* Gv29-8 [12]. The GO analysis found 53 new terms, which were put into three groups (Appendix A), including cellular components, biological processes, and molecular functions. Among them, the protein-containing complex, metabolic process, and catalytic activity showed the largest number of GO terms in different periods (9 d vs. 3 d and 12 d vs. 3 d, respectively) (Appendix A). These results are consistent with the results studied by Yuan et al. [58]. Similarly, GO enrichment analyses of DEGs have been shown at the different growth stages of *T. harzianum* in the presence of BCCW [59]. This fact is consistent with the extensive metabolic activity expected for *Trichoderma* during the mycoparasitism mechanism [60]. Usually, annotations provide a valuable resource for specific processes, functions, and pathways in species of *Trichoderma* [61].

A total of nine pathways were also categorized with different expression levels by KEGG analysis under four categories: metabolism, environmental information processing, the cellular system, and human diseases. The diagrams on Figure 10 showed the largest number of KEGG pathways in different periods (9 d vs. 3 d and 12 d vs. 3 d, respectively). KEGG is working on classifying differentially expressed genes. Recently, KEGG analysis assigned 2320 genes to 87 pathways, the proteasome and ribosome pathways being the most significantly enriched in the *T. longibrachiatum* [61]. The important pathway was the glyoxylate and dicarboxylate metabolisms, which have demonstrated a clear relationship with the metabolism functions involved in the biocontrol mechanism [60].

The six differentially expressed genes were found to be related to microsclerotia degrading enzymes of *T. virens* HZA14. These enzymes include endochitinase A1, endochitinase 3, and alpha-N-acetylglucosaminidase and glucanases endo-1,3-beta-glucanase, laccase-1, and peroxidase. These enzymes were mainly released by *Trichoderma* species during the mycoparasitism process to penetrate the fungal cell wall [62]. The endochitinase and endo-1,3-beta-glucanase enzymes acted on cell-wall degradation through cleaving ß-linkages internally in both ß-glucan and chitin and their oligomers at random along the polysaccharide chain [63]. Li et al. [64] found that the mycoparasitism of *T. atroviride* is linked to the enzyme chitinase, which breaks down the cell wall of the aeciospores of *Cronartium ribicola*, while protease enzymes attack specific amino acid residues within the polypeptide chain [62]. Melanin in the microsclerotia is composed of the oxidative polymerization of various phenolic compounds, which are degraded by phenol-oxidases such as laccase. A few studies indicate that the laccase gene *Lcc1* is specifically highly expressed in *T. virens* during the mycoparasitic process against the sclerotia of *B. cinerea* [65,66]. These enzymes act synergistically in a mycoparasitic process, resulting in the weakening and hydrolysis of cell walls of fungal hosts and allowing for easier access of *Trichoderma* to nutrients [57].

In this study, both transcriptome data and RT-qPCR were used to analyze and confirm the enzyme-coding genes related to microsclerotial degradation. The results showed that the expression of six enzyme-coding genes were all up-regulated after 3, 6, 9, 12, and 15 days and differed in different enzyme-coding genes at different stages of interactions between the hyphae of *T. virens* HZA14 and the microsclerotia of *V. dahliae* (Figure 11). Similarly, RT-qPCR confirmed the expression patterns of six enzyme-coding genes in the same transcriptome samples, showing that transcriptome data is accurate (Figure 12). In this study, the phenomenon of microsclerotial degradation by *T. virens* was unveiled for the first time. This process involves the interaction of fungal hyphae, initiating the release of various enzymes to facilitate the invasion and degradation of solid microsclerotia by *Trichoderma* hyphae [54,67]. While it is evident that cell wall-degrading enzymes play a pivotal role in this mechanism, the precise temporal sequence in which these enzymes are deployed remains somewhat unclear based solely on transcriptome data.

## 4. Materials and Methods

### 4.1. Fungal Materials

Fifteen *Trichoderma* isolates stored at −20 °C in a 10% glycerol solution were used, including *T. atroviride* HZA1, *T. atroviride* HZA2, *T. afroharzianum* HZA3, *T. brevicompactum* HZA4, *T. dorothopsis* HZA5, *T. koningiopsis* HZA6, *T. brevicompactum* HZA7, *T. dorothopsis* HZA8, *T. citrinoviride* HZA9, *T. asperellum* HZA10, *T. harzianum* HZA11, *T. brevicompactum* HZA12, *T. atroviride* HZA13, *T. virens* HZA14, and *T. dorothopsis* HZA15. The high aggressive phytopathogenic isolate H6 of *V. dahliae* was obtained from the Plant Pathology Department, Agriculture and Biotechnology College, Zhejiang University, Hangzhou, China [68]. All the fungal isolates were cultured on the plates containing PDA at 25 °C for five days for *Trichoderma* isolates and 14 days for *V. dahliae* isolate.

### 4.2. Screening of Trichoderma Isolates with Antagonistic Activity against V. dahliae

For determining the *Trichoderma* isolate with high antagonistic activity against *V. dahliae* in the PDA medium, fifteen *Trichoderma* isolates were screened by a dual culture method, as described by [57] with slight modifications. In brief, mycelial discs (0.5 cm diameter) from *V. dahliae* isolate grown for 14 days on PDA at 20 °C were transferred to one side of a plate containing PDA, and after the plate was incubated for three days, another mycelial disc (5 mm diameter) from *Trichoderma* isolate grown on PDA at 25 °C for seven days was placed 1.5 cm apart on the same PDA plate. All plates were incubated at 23 °C in darkness for 15 days, and each treatment for each *Trichoderma* isolate was replicated three times. The growths of the pathogen *V. dahliae* and *Trichoderma* isolates were observed, and their antagonistic activities were evaluated according to the antagonism scale from class 1 to class 5 [69]: Class 1: *Trichoderma* grew on the entire surface of the medium and covered the pathogen; Class 2: *Trichoderma* grew on 2/3 of the medium, and the pathogen grew on the last third; Class 3: *Trichoderma* grew on 1/2 of the medium, and the pathogen grew on the second half; Class 4: the pathogen grew on 2/3 of the medium, and *Trichoderma* grew on the last third; Class 5: the pathogen grew on the whole medium and covered the *Trichoderma*.

### 4.3. Culture Filtrate Activity Produced by Trichoderma Isolates against V. dahliae

To evaluate the bioactivity of metabolites produced by 15 isolates of *Trichoderma* against *V. dahliae*, the six 0.5 cm diameter discs for each *Trichoderma* isolate grown in PDA at 25 °C for five days were placed into a 250-mL conical flask containing PD broth of 100 mL, and the flasks were incubated in a ZWY-211B rotary shaker at 150 rpm at 25 °C for 14 days. After incubation, the partial liquid was filtered using cheesecloth, centrifuged at 6000 rpm for 5 min at 4 °C, and the supernatant was filtered through Millipore filter paper of 0.22 μm diameter. The bioactivity of culture filtrate against the hyphal growth of the pathogen *V. dahliae* was evaluated using the poisoned food method. The 1.0 mL of culture filtrate with concentrations of 0, 25%, 50%, and 100% were mixed with 9 mL of PDA in a plate. After medium solidification, a 5 mm mycelial disc from a *V. dahliae* isolate grown for 14 days was placed onto the center of a plate containing PDA and incubated at 23 °C for 10 days. To determine the extent of inhibition of radial mycelial growth, a formula [(V − T)/V] × 100 was employed, which involved calculating the percentage difference between the growth observed in the control (V) and the growth observed in the treatment (T) [70]. The well diffusion assay was performed for the evaluation of culture filtrate in the inhibition of the conidia germination of *V. dahliae*. In brief, 0.5 mL of fungal spore suspension (10^6^ conidia/mL) was spread onto the surface of PDA in a plate after drying in a fume hood. Wells of 0.3 cm diameter were prepared in the center of the plates using a sterilized stainless steel cork borer. The 30 µL of culture filtrates at 25%, 50%, and 100% were added to a well, respectively, and incubated at 23 °C for three days. The diameters of inhibition zones around the wells were measured. Each treatment had three repeats, and no culture filtrate treatment was used as a control. The data were analyzed using analysis of variance (ANOVA) and the LSD test at *p* < 0.05.

### 4.4. Detection of Siderophore and 3-Indoleacetic Acid (IAA) Production

In order to determine the siderophore production of *Trichoderma* isolate, a chrome azurol sulphonate (CAS) agar plate as described by Milagres et al. [71] with slight modifications was used. To prepare the CAS-blue agar medium, a series of steps were followed. Initially, 60.5 mg of CAS was dissolved in 50 mL of distilled deionized water. Then, 10 mL of iron (III) solution (consisting of 1 mM FeCl·6H_2_O and 10 mM HCl) was added to the CAS solution. This mixture was slowly added to 72.9 mg of HDTMA, which was previously dissolved in 40 mL of water and stirred well. The resulting solution was dark blue and was autoclaved at 121 °C for 25 min. In a separate container, another mixture was formed by combining 750 mL of water, 15 g of agar, 30.24 g of pipes, and 12 g of a 50% (*w*/*w*) NaOH solution to adjust the pH to 6.8. Finally, the dye solution was carefully poured along the glass wall of the container and agitated gently to avoid foaming [72]. Petri dish plates of 10 cm divided into halves were prepared; the PDA was poured in the first half, and CAS-blue agar was poured in the second half to be at the same level as the first half. A disc (0.5 cm diameter) from *Trichoderma* isolate grown for five days was inoculated onto the center of the PDA side. The plates were incubated at 27 °C. The color change was monitored in the second half after 14 days.

In order to assay the IAA production of *Trichoderma* isolate, the Salkowski reagent was used as described by Glickmann and Dessaux [73]. Five mycelial discs (5 mm) from the 4-day-old colony were put in a conical flask comprised of 100 mL PD broth added with 1.5% of L-tryptophan, while treatment of PD broth was used as the control and then incubated in a ZWY-211B rotary shaker at 150 rpm at 25 °C in the darkness for 5, 10, and 15 days, respectively. The liquid part was filtered using cheesecloth and centrifuged at 6000 rpm for 5 min at 4 °C, and then the supernatant was filtered through Millipore filter paper of 0.22 μm diameters. The supernatant was adjusted to pH 7 using 1 N NaOH. To determine IAA production, the Salkowski reagent was prepared (150 mL of HClO_4_, 250 mL of ddH_2_O, and 7.5 mL of 0.5 M FeCl_3_·6H_2_O), and 2 mL from the Salkowski reagent solution was mixed with 1 mL bacterial supernatant in a test tube [74]. The tubes were incubated in darkness at 25 °C for 30 min and observed for the development of pink color compared with the control. The absorbance was measured at a wavelength of 530 nm using a UV–Vis spectrophotometer (Perkin Elmer Lambda 35, Waltham, MA, USA). A standard curve was made from 1–100 µg/mL of commercial IAA (Sigma-Aldrich, Chemie GmbH. Eschenstr. 5. 82024 Taufkirchen. Germany) solution in PD broth for a measure of the concentration of IAA in the *Trichoderma* isolate supernatant as µg/mL. The experiment was carried out using three replicates; the data were analyzed using analysis of variance (ANOVA). The LSD test at *p* < 0.05 was applied to compare the means with a confidence interval of 95%.

### 4.5. Preparation of Fungal Inoculations

The inoculum of *Trichoderma* isolate was prepared with culture in flasks containing boiled wheat grains autoclaved two times at 121 °C for 2 h. The flasks were incubated at 25 °C with a 12/12 h photoperiod for 15 days. The conidia suspension was prepared by putting grains with conidia into sterile water, shaking it well, and filtrating it with four layers of sterile cheesecloth [75]. The concentration of conidia suspension was adjusted using the hemacytometer. For preparing *V. dahliae* microsclerotia, the discs with pathogen hyphae were put onto a plate containing a sterile, modified basal agar medium covered by a sterile cellophane membrane. A total of 20 plates were prepared, and they were incubated at 20 °C for 25 days in the darkness [76]. The black microsclerotia were scraped using the sterile blade and were added to 200 mL tap water before being mixed by a blender. The resulting suspension was purified using a series of 125, 74, and 20 μm sieves [77]. The microsclerotia from the 20 μm sieve were washed two times in tap water, transferred to sterile water, and centrifuged at 4000 rpm for 5 min. To kill mycelia and conidia, the tubes with microsclerotia were put in the water bath at 47 °C for 5 min [78]. Microsclerotia were suspended in sterile water and counted using a hemacytometer.

### 4.6. Greenhouse Experiment

In order to evaluate the effectiveness of *Trichoderma* in reducing the *Verticillium* wilt disease caused by phytopathogenic *V. dahliae*, the soil of pots was inoculated with the inoculum of *T. virens* HZA14 and microsclerotia of *V. dahliae* before planting eggplant plants at different time periods. The field soil mixed with peat, vermiculite, and sand (ratio of 3:1:1, respectively) was autoclaved, and conidia of *T. virens* HZA14 and microsclerotia of *V. dahliae* were prepared. The sterile soil was mixed with the microsclerotia and conidia 10, 20, and 30 days before sowing with final concentrations 1 × 10^7^ conidia g^−1^ for *Trichoderma* isolates and 20 microsclerotia g^−1^ for *V. dahliae* in soil [79]. No microsclerotial soil was used as a control. The pots were filled with the soil, and a susceptible variety of eggplant plant seeds from the Hangzhou Academy of Agricultural Science (cv. Zheqie-3) were washed many times with sterile water after sterilizing with a 2% Clorox solution for 2–3 min. The seeds were then planted into pots and subjected to four different periods before being moved under controlled conditions of 75% humidity and temperatures ranging from 22–28 °C. Thirty plants (*n* = 30) were used for each treatment, and three independent replicates were performed for each treatment. Disease severity and control efficacy were assessed when the death of some plants appeared in the control treatment [80]. Assessment of disease severity used the following rating scale: 0 = no diseased leaf; 1 = <10%; 3 = 11–25%; 5 = 26–50%; 7 = >50%; 9 = plant killed [80].
$$Disease\;index=\sum \frac{\left(number\;of\;leaves\;with\;\mathrm{every}\;\mathrm{scale}\times disease\;severity\;score\right)}{\left(the\;total\;number\;of\;leaves\;examined\;\times the\;highest\;severity\;score\right)}\times 100\%$$
$$Control\;efficacy\;=\frac{\left(the\;mean\;disease\;index\;of\;control\;-\;the\;mean\;disease\;index\;of\;treatment\right)}{\left(the\;mean\;disease\;index\;of\;the\;control\right)}\times \;100\%$$

Statistical analyses by one-way analysis of variance (ANOVA) were done using SPSS software version 16 (SPSS, Chicago, IL, USA), and the significance level for the LSD test was set at *p* < 0.05 with a confidence interval of 95%.

### 4.7. Plant Growth Promotion (PGP) Assay

To study the potency of the *Trichoderma* isolate in improving the growth of eggplant seedlings, seeds of eggplants (cv. Zheqie-3) were sown into the pots filled with the mixed soil (as detailed above) with 1 × 10^7^ conidia g^−1^ of the *Trichoderma* isolate. No micro-sclerotial soil was used as the control. Experimental treatments (twenty pots) were done in triplicate. The pots were returned to the same controlled conditions as detailed above. The PGP potential of isolate HZA14 was assessed by measuring the seedling height, the length of the taproots, and fresh and dry weights at 30 days after sowing. The dry weights were calculated after seedling organs were dried for 2 days at 60 °C. The formula PGPE% = [(treatment − control)/control] × 100% was followed to calculate growth promotion efficacy. Statistical analyses are described as detailed in Section 4.6.

### 4.8. RNA Extraction and Purification

To determine the genes involved in the degradation of microsclerotia of *V. dahliae* during the mycoparasitism of *T. virens* HZA14, transcriptome analysis was performed following the dual culture of the two fungi. A mycelial disc of *V. dahliae* isolate was transferred to one side of a plate containing PDA, and after the plate was incubated for three days, another mycelial disc from a selected *Trichoderma* isolate grown on PDA at 25 °C was placed 1.5 cm apart on the same PDA plate. All plates were incubated at 23 °C in darkness, and then the interaction zones between the hyphae of *Trichoderma* isolate and the microsclerotia of *V. dahliae* were cut 3, 6, 9, 12, and 15 days after incubation, respectively. Each treatment had three replicates during each period of time. The harvested interaction zones were frozen in liquid nitrogen immediately. The Trizol1 method was used to extract total RNA, as described by the manufacturer. RNA purity was checked using the kaiaoK5500^®^ Spectrophotometer (Kaiao, Beijing, China). The RNA was frozen in liquid nitrogen for 1 h, stored at ×80 °C, and delivered to Zhejiang Annoroad Biotech Ltd. for transcriptome sequencing.

### 4.9. Library Preparation for RNA Sequencing

To prepare the RNA sequencing libraries extracted from *T. virens* HZA14, 2 μg of RNA were used as input material for each sample. The NEBNext^®^ Ultra™ RNA Library Prep Kit for Illumina^®^ (#E7530L, NEB, USA) was used to generate sequencing libraries following the manufacturer’s instructions, with index codes added to attribute sequences to each sample. The mRNA was isolated from total RNA using poly-T oligo-attached magnetic beads, and fragmentation was carried out using divalent cations under high temperature in NEBNext First Strand Synthesis Reaction Buffer (5×). The first strand of double-stranded cDNA was synthesized via RNase H and random hexamer primer. The second strand of the cDNA was synthesized using dNTPs, RNase H, DNA polymerase I, and buffer. The library fragments were purified using QiaQuick PCR kits and eluted with EB buffer. Terminal repair, A-tailing, and adapter addition were performed, and the aimed products were retrieved. PCR amplification was performed to complete the library. Qubit^®^ RNA Assay Kit in Qubit^®^ 3.0 was then used to determine the RNA concentration of the library; after that, RNA was diluted to 1 ng/μL. An Agilent Bioanalyzer 2100 system (Agilent Technologies, Santa Clara, CA, USA) was used to measure the insert size, and then StepOnePlus™ Real-Time PCR System (library valid concentration > 10 nM) was used for precisely measuring the qualified insert size. The clustering of the index-coded samples was carried out on a cobalt cluster generation system using the HiSeq PE Cluster Kit v4-cBot-HS (Illumina, San Diego, CA, USA) according to the manufacturer’s instructions. After cluster generation, the libraries were sequenced on an Illumina platform with 150 bp paired-end reads.

### 4.10. Filter of Original Data

To ensure the accuracy and reliability of the transcriptome data, a Perl script was utilized to filter the raw data. The script removed adapter sequences, poly-N, and low-quality bases from the raw data to obtain clean reads. Quality control measures such as Q20, Q30, GC-content, and sequence duplication level were also calculated for the processed data. Reference genomes, annotation files, and regulatory functions were downloaded from the ensemble database (http://www.ensembl.org/index.html, accessed on 5 July 2021). The genome index was built using Bowtie2 v2.2.3, and the clean data were aligned to the reference genome using HISAT2 v2.1.0 [81]. The mapping results were visualized using the Integrative Genomics Viewer (IGV) and displayed as a heatmap [82]. HTSeq v0.6.0 was used to count the reads for each gene in each sample, and the correlation between samples was determined [83]. The expression level of each gene in each sample was estimated using fragments per kilobase million mapped reads (FPKM), which was calculated with Cufflinks v2.0.2. The formula used for calculating FPKM is FPKM = [106 × F/(NL/103)], where F represents the number of fragments assigned to a certain gene, N represents the total number of mapped reads, and L represents the length of the gene [84].

### 4.11. Analysis of Differential Gene Expression

In order to analyze differential gene expression between two biological samples with replicates, the statistical tool DESeq2 was used. This technique is based on the negative binomial distribution theory, which assumes that count values follow this distribution. DESeq2 estimates sample-specific and gene-specific depth parameters and uses “linear regression” for dispersion to “shrink” the variance of gene expression, taking into account the similarity of gene expression levels. It then estimates the expression level of each gene in each sample using linear regression and calculates the *p*-value with the Wald test. Finally, the p-value is corrected by the BH method. Genes with q ≤ 0.05 and |log2_ratio| ≥ 1 are considered DEGs.

### 4.12. Analysis of Function Enrichment of Gene Ontology

GO enrichment analysis was used with the hypergeometric test to reveal the biological functions of DEGs. This helped us figure out the possible biological processes and molecular functions linked to the DEGs. The *p*-value was calculated and adjusted to the q-value, with the data background being the genes in the whole genome. GO terms with q-values less than 0.05 were considered significantly enriched. To identify significantly enriched pathways related to the DEGs, the KEGG enrichment analysis was also performed. The *p*-value was adjusted by multiple comparisons to obtain the q-value. KEGG terms with q-value < 0.05 were considered as significantly enriched. This analysis provided insight into the metabolic pathways and biological functions associated with the DEGs.

### 4.13. Quantitative Reverse Transcription PCR (qRT-PCR)

The cell walls in filamentous fungi are mainly composed of glucan, chitin, and proteins that are extensively cross-linked to form a complex network, which forms the structural basis of the cell wall. In light of that, we took into account six of the main enzyme-coding genes potentially involved in the degradation of microsclerotia, including enzymes endochitinase A1, endochitinase 3, endo-1,3-beta-glucanase, alpha-N-acetylglucosaminidase, laccase-1, and peroxidase. All positive matched hits with an e-value ≥ 1 × 10^−15^ and coverage less than 0.5 were examined manually for each enzyme in the genome based on the active domains in the enzyme database [85]. Statistical analyses of triplicates for each gene were done using SPSS software version 16 (SPSS, Chicago, IL, USA), and the significance level for the LSD test was set at *p* < 0.05. Confidence intervals of 95% were set. To confirm the DEGs results obtained, RT-qPCR experiments were carried out [86]. The interaction zones between the hyphae of *Trichoderma* isolate and the microsclerotia of *V. dahliae* were cut 3, 6, 9, 12, and 15 days after incubation, respectively, as described above. Total RNAs from the five different time periods’ samples were extracted by using TRIZOL^®^ reagent (TaKaRa RNAiso Reagent 9180). Reverse transcription of total RNA to cDNA was carried out by using a protocol of the TAKARA fluorescence quantitation reagent (TB Green Premix Ex Taq™ RR420A) and TAKARA reverse transcription reagent (PrimeScript™ RT reagent Kit with gDNA Eraser RR047A) (TaKaRa, Otsu, Japan) (Appendix A). Six of the main genes potentially involved in the degradation of microsclerotia were amplified by RT-qPCR using the primer pairs designed by the PerlPrimer v1.1.20 software (Table 4), and the actin gene was used as an internal reference gene to normalize gene expression.

The 25 μL of reaction system was composed of TB Green Premix Ex Taq TM II (Tli RNaseH Plus), both forward and reverse primers, the cDNA template, and nuclease-free water, as showed in Appendix A.

Reaction programs were 95 °C pre-denaturation for 30 s, 95 °C for 5 s, 60 °C for 30 s for 45 cycles, 95 °C for 15 s, 60 °C for 60 s, and 95 °C for 15 s. Fluorescence quantification was assessed using a Takara kit and the instrument Bio-Rad CFX96 Real-Time System (Bio-Rad, Hercules, CA, USA). The results of gene expression levels were calculated and statistically analyzed using the threshold cycle (2^−ΔΔCt^) method, and the data were expressed as means ± standard deviations. Statistical analyses were done using SPSS software version 16 (SPSS, Chicago, IL, USA), and the significance level for the LSD test was set at *p* < 0.05. Confidence intervals of 95% were set.

## 5. Conclusions

Microsclerotia, serving as the primary source of inoculation for numerous hosts, represent the primary survival and dormancy structures of the fungal pathogen *V. dahliae*. Our study has confirmed that microsclerotial degradation by *T. virens* HZA14 is a crucial mechanism in controlling *Verticillium* wilt. Furthermore, we have observed that *T. virens* HZA14 possesses the capacity to produce siderophores and IAA. Transcriptome analysis showed the gene expression profiles of *T. virens* HZA14 in the process of microsclerotial degradation. The DEGs were identified, and *Trichoderma* genes related to microsclerotial degradation were characterized, as supported by both transcriptome data and RT-qPCR validation. Finally, the transcriptome data suggest the involvement of candidate genes (endochitinase A1, endochitinase 3, endo-1,3-beta-glucanas, alpha-N-acetylglucosaminidase, laccase-1, and peroxidase) in mycoparasitism in *T. virens* HZA14.

## Figures and Tables

**Figure 1 plants-12-03761-f001:**
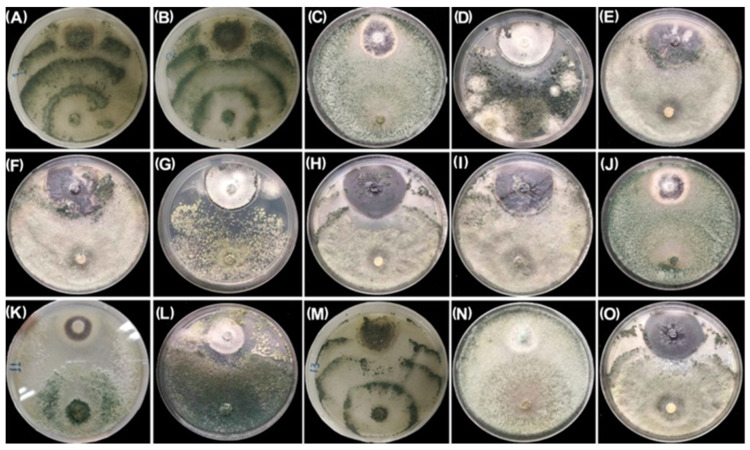
Antagonistic activity of fifteen *Trichoderma* isolates against *V. dahliae* on PDA at 23 °C for 15 days. (**A**) *T. atroviride* HZA1. (**B**) *T. atroviride* HZA2. (**C**) *T. afroharzianum* HZA3. (**D**) *T. brevicompactum* HZA4. (**E**) *T. dorothopsis* HZA5. (**F**) *T. koningiopsis* HZA6. (**G**) *T. brevicompactum* HZA7. (**H**) *T. dorothopsis* HZA8. (**I**) *T. citrinoviride* HZA9. (**J**) *T. asperellum* HZA10. (**K**) *T. harzianum* HZA11. (**L**) *T. brevicompactum* HZA12. (**M**) *T. atroviride* HZA13. (**N**) *T. virens* HZA14. (**O**) *T. dorothopsis* HZA15.

**Figure 2 plants-12-03761-f002:**
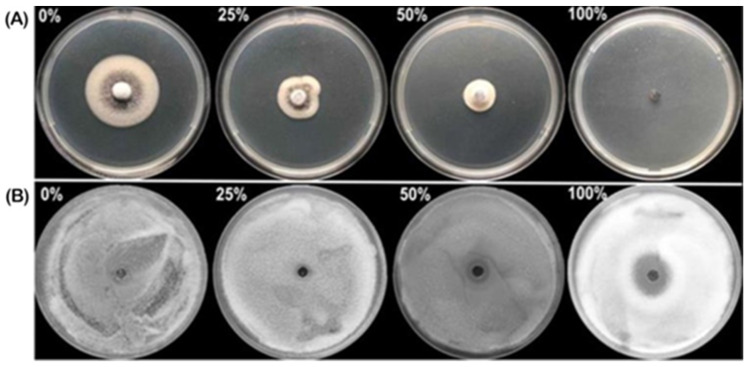
The activities of culture filtrate produced by *T. virens* HZA14 against *V. dahliae*. (**A**) Percent inhibition of mycelial growth of *V. dahliae* at concentrations of 0, 25%, 50%, and 100%. (**B**) Percent inhibition of conidia germination of *V. dahliae* at concentrations of 0, 25%, 50%, and 100%.

**Figure 3 plants-12-03761-f003:**
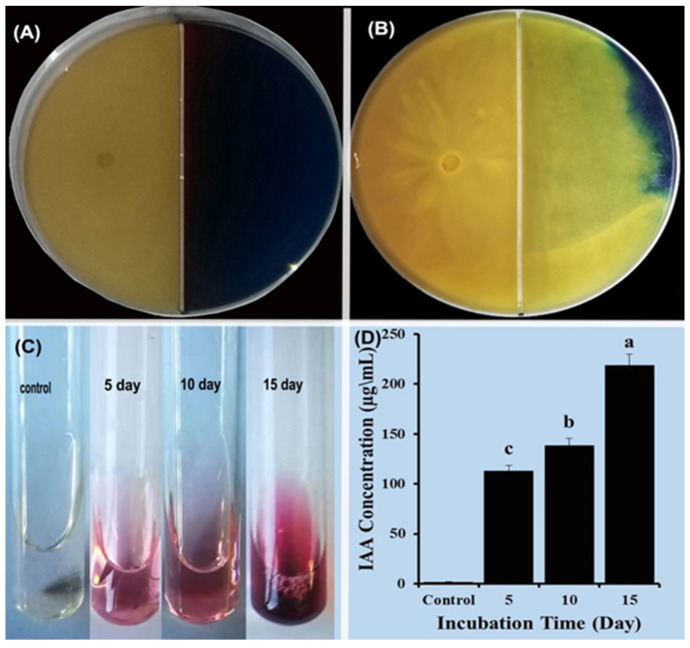
Siderophore and IAA produced by isolate *T. virens* HZA14. (**A**) Growth of isolate HZA14 for three days. (**B**) Hyphae of *T. virens* isolate HZA14 growing on CAS medium after 11 days. (**C**) Color changes after the mixture of culture supernatants with Salkowski’s reagent. (**D**) Quantification of IAA. Data with different letters are considered significantly different (*p* ≤ 0.05).

**Figure 4 plants-12-03761-f004:**
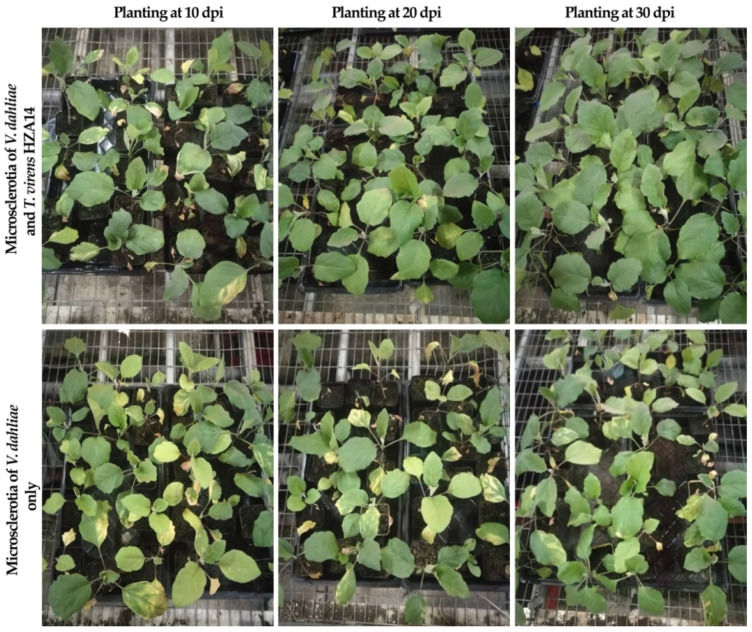
The potential of *T. virens* isolate HZA14 in reducing *Verticillium* wilt on eggplant seedlings.

**Figure 5 plants-12-03761-f005:**
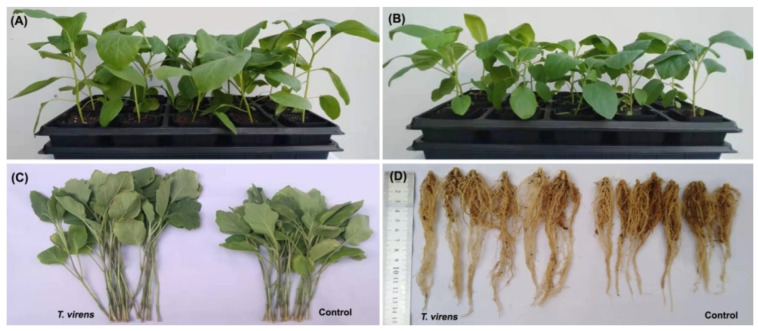
Plant growth promotion ability of *T. virens* isolate HZA14 against eggplant seedlings 30 days after sowing. (**A**,**B**) Inoculation with *T. virens* isolate HZA14 and without (Control). (**C**) Stem length of eggplant seedlings inoculated with *T. virens* isolate HZA14 and without. (**D**) Root length of eggplant seedlings inoculated with *T. virens* isolate HZA14 and without.

**Figure 6 plants-12-03761-f006:**
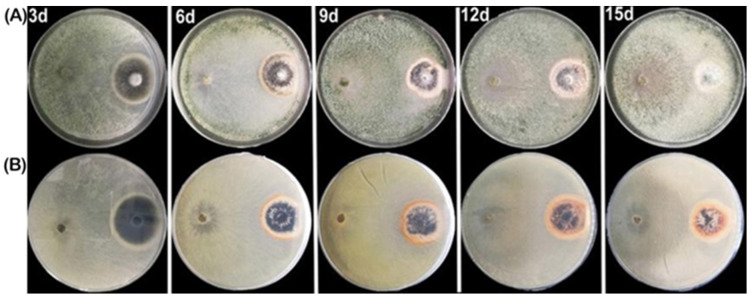
Process of microsclerotial degradation of *V. dahliae* by *T. virens* HZA14 during the mycoparasitism on PDA medium. (**A**) The front of petri dishes. (**B**) The reverse of petri dishes.

**Figure 7 plants-12-03761-f007:**
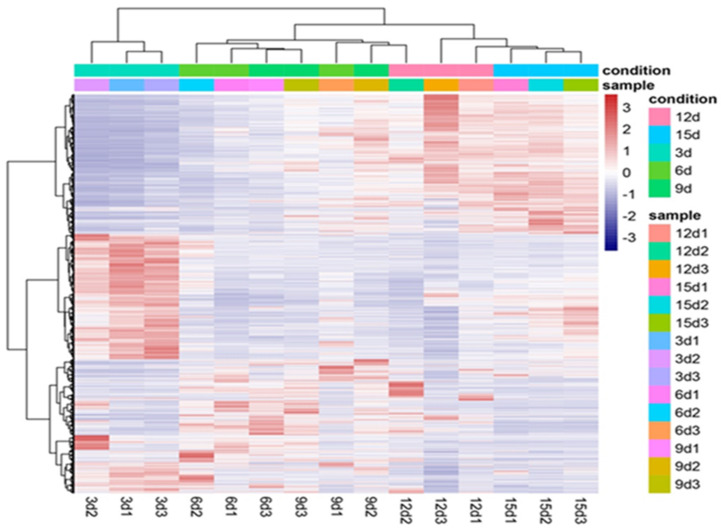
Heatmap showing gene expression in interaction between *T. virens* HZA14 and *V. dahliae* in five different periods of time. The X-axis shows different time treatments, and the Y-axis shows expressed genes. Red indicates high expression levels, and blue indicates low expression levels. Data for gene expression level were normalized to Z-scores.

**Figure 8 plants-12-03761-f008:**
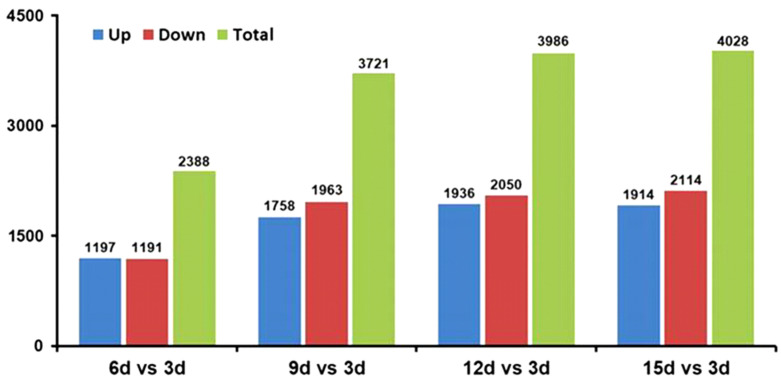
Comparison of the identified differentially expressed genes (DEGs) of *T. virens* HZA14 during the mycoparasitism on the *V. dahliae* within different times periods.

**Figure 9 plants-12-03761-f009:**
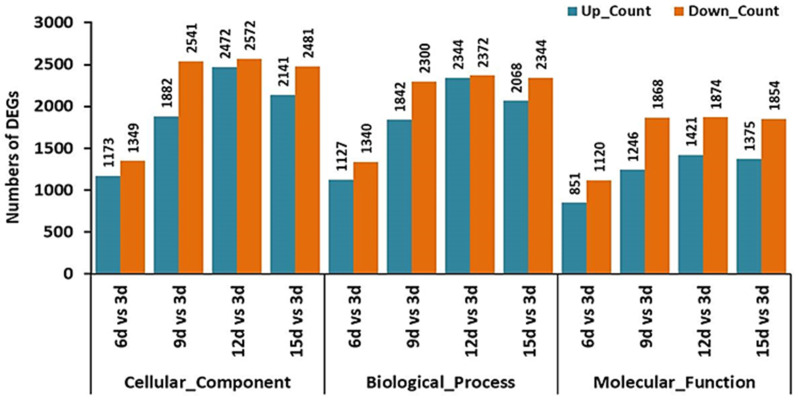
Gene ontology classification of differentially expressed genes (DEGs) in 15 samples.

**Figure 10 plants-12-03761-f010:**
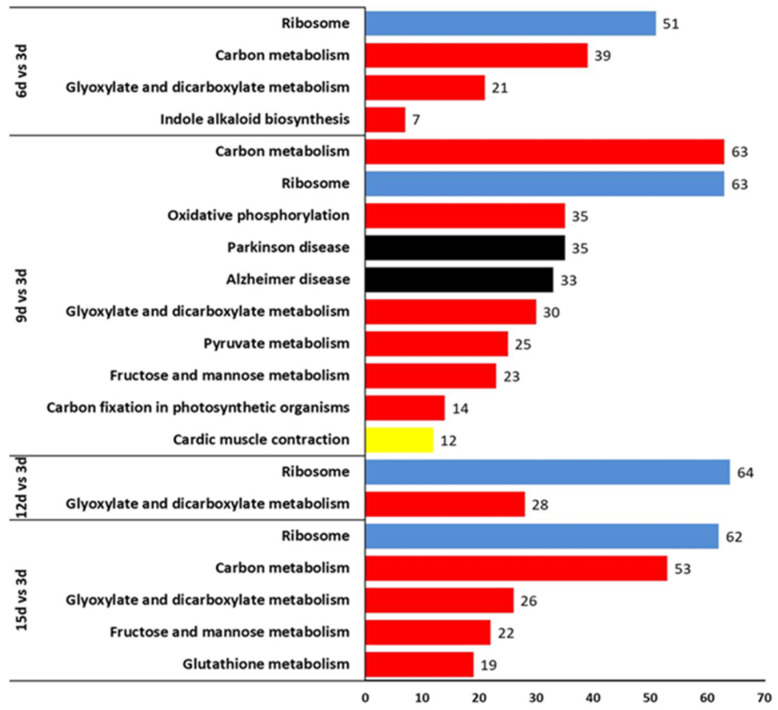
KEGG pathway enrichment analysis, the significant pathways for differentially expressed genes for time periods 6 d vs. 3 d, 9 d vs. 3 d, 12 d vs. 3 d, and 15 d vs. 3 d. Genes were divided into four categories according to the involvement of the KEGG pathway. Blue, Environmental information processing; Red, Metabolism; Black, Human diseases; Yellow, Organismal system.

**Figure 11 plants-12-03761-f011:**
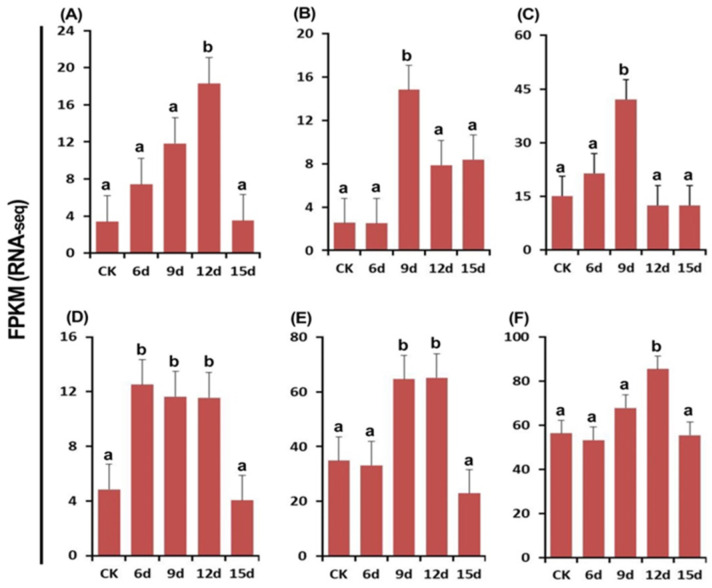
Analysis of enzyme-coding DEGs related to enzymes of microsclerotial degradation of *V. dahliae* after 3, 6, 9, 12, and 15 days of incubation. (**A**) Endochitinase A1. (**B**) Endochitinase 3. (**C**) Endo-1,3-beta-glucanase. (**D**) Alpha-N-acetylglucosaminidase. (**E**) Laccase-1. (**F**) Peroxidase. The columns represent the mean FPKM for three replicates. Data with different letters are considered significantly different (*p* ≤ 0.05).

**Figure 12 plants-12-03761-f012:**
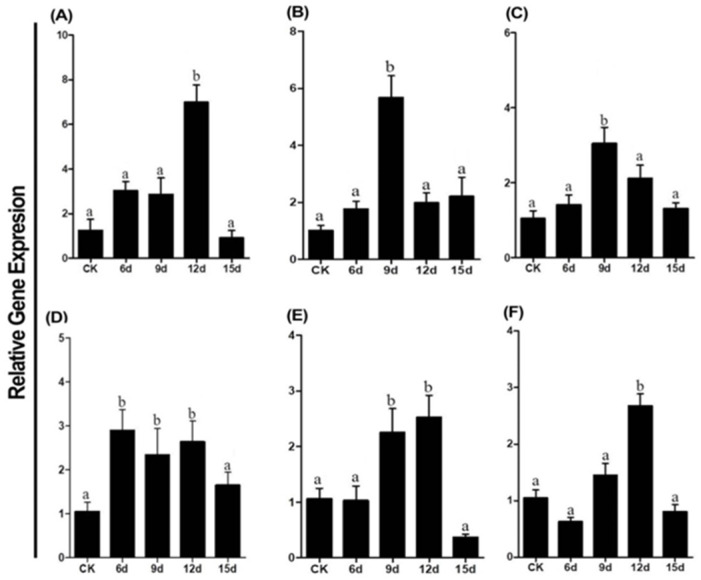
Expression levels of six genes in RT-qPCR detection after 3, 6, 9, 12, and 15 days of incubation. (**A**) Endochitinase A1. (**B**) Endochitinase 3. (**C**) Endo-1,3-beta-glucanase. (**D**) Alpha-N-acetylglucosaminidase. (**E**) Laccase-1. (**F**) Peroxidase. The columns represent the mean relative gene expression for three replicates. Data with different letters are considered significantly different (*p* ≤ 0.05).

**Table 1 plants-12-03761-t001:** Effect of culture filtrates produced by *T. virens* HZA14 on mycelial growth and conidia germination of *V. dahliae*.

Concentration of CF ^1^	PIRMG ^2^ (%)	IZD ^3^ (mm)
100%	100.00 ± 0.0 ^a^	10.13 ± 0.3 ^a^
50%	65.51 ± 8.9 ^b^	3.49 ± 0.1 ^b^
25%	50.71 ± 2.9 ^c^	1.48 ± 0.1 ^c^
0	----------	0.00 ± 0.0 ^d^

^1^ CF: culture filtrate; ^2^ PIRMG: Percentage inhibition of radial mycelial growth; ^3^ IZD: Inhibition zone diameters. Value are means ± standard deviations of three replicates, and the different lowercase letters in the same column are significantly different at *p* < 0.05 according to the LSD test.

**Table 2 plants-12-03761-t002:** Control efficacy of *T. virens* isolate HZA14 against *Verticillium* wilt of eggplant seedlings.

Periods/Days	Disease Severity (%)		Control Efficacy (%)
	*V. dahliae* Alone	*T. virens* + *V. dahliae*	
10	72.96 ± 2.62 ^a^	44.43 ± 0.95 ^c^	39.10 ± 1.30 ^c^
20	74.63 ± 0.96 ^a^	23.88 ± 0.62 ^b^	67.99 ± 0.83 ^b^
30	71.48 ± 1.91 ^a^	2.77 ± 0.62 ^a^	96.59 ± 0.76 ^a^
Average	75.69	23.69	67.89

Values are means ± standard deviations of three replicates, and the different lowercase letters in the same column are significantly different at *p* < 0.05 according to the LSD test.

**Table 3 plants-12-03761-t003:** Percent differences of various growth parameters between treatment and control 30 days after inoculation with *T. virens* HZA14.

Treatments	Measured Parameters					
	Stem Length (cm)	Stem Fresh Weight (g)	Stem Dry Weight (g)	Root Length (cm)	Root Fresh Weight (g)	Root Dry Weight (g)
HZA14	15.58 ± 0.47 ^a^	8.04 ± 0.39 ^a^	0.77 ± 0.01 ^a^	14.29 ± 0.26 ^a^	0.64 ± 0.029 ^a^	0.17 ± 0.013 ^a^
Control	13.37 ± 0.43 ^b^	6.86 ± 0.38 ^b^	0.64 ± 0.02 ^b^	12.59 ± 0.21 ^b^	0.50 ± 0.022 ^b^	0.11 ± 0.011 ^b^
GPE (%)	16.54	17.20	20.31	13.50	28.00	54.55

Values are means ± standard deviations of three replicates, and the different lowercase letters in the same column are significantly different at *p* < 0.05 according to the LSD test. GPE: growth promotion efficacy.

**Table 4 plants-12-03761-t004:** Primers utilized in qPCR experiments.

Gene Name	Forward and Reverse of Primers (5′ to 3′)
*Endo-1,3-beta-glucanase*	F: CACACCACCGTCCTCAAGGGCTCCG R: GTGGGTGAATCGGGCGACAATGAGA
*Endochitinase A1*	F: ACCCCGTAACTGGCTTGCCCACACA R:TGGAAGGGAAGAGAGTAGAGTTGCT
*Endochitinase 3*	F: CTACCCTCCGTCCCTTTGGCACTGT R: GGCGTCGGGAAAGGGGCACTGGGGA
*Alpha-N-acetylglucosaminidase*	F: ATTCGTCCCCCGCAACATCTCTCGC R: CCACTGAGGAGCGGATTCGGCAAAC
*Laccase-1*	F: TGAGGGGCACGAGGACGATGATGAG R: GCGTTAGGATAAAATAGCAGAGGGT
*Peroxidase*	F: TCCGACGCCTGGGACGCCCTCACTG R: CGAAGTAGCGGAAGCCCTGGGTGTC
*Actin*	F: GGTCAGGTCATCACCATCGGCAACG R: GCTGCTGTGTGAATGGATGGGAAAA

## Data Availability

The data are contained within the manuscript and Appendix A.

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
