# Peer review of "Potential of Trichoderma virens HZA14 in Controlling Verticillium Wilt Disease of Eggplant and Analysis of Its Genes Responsible for Microsclerotial Degradation"

_plants, 2023, doi:10.3390/plants12213761_

Round 1

Reviewer 1 Report

Comments and Suggestions for Authors

Dear author(s),

The following points should clearly be corrected and explained for readers in the manuscript:

Abstract

1.       L17, write a start sentence about importance of topic/study.

2.       L23, remove -

3.       L31, a conclusion sentence should be written for readers.

Keywords

4.        Scientific name of eggplant should be written.

Introduction

5.       L35, author name of Verticillium dahlia should be written in the first mentioned place.

6.       L37, scientific name of eggplant and its author name should be written in the first mentioned place.

7.       L38, remove “… in Zhejiang province” because this disease is widespread in the world.

8.       L56, change Transcriptome to transcriptome

9.       L59, T. virens should be italicized.

10.    L60, author name of Rhizoctonia solani should be written in the first mentioned place.

11.    L63, author name of Pythium ultimum should be written in the first mentioned place

12.     L65, Rhizopus oryzae should be italicized.

13.    L65, author name of Rhizopus oryzae should be written in the first mentioned place

14.    L67, author name of Sclerotinia sclerotiorum should be written in the first mentioned place

15.    L67, author name of Botrytis cinerea should be written in the first mentioned place

16.    L68, a blank after dot.

17.    L69, author name of T. harzianum should be written in the first mentioned place

18.    L70-71, enzymes are not special name and they should be written with lower case letter.

19.    L75, change Proteins to proteins

20.    L78, Verticillium could be italicized.

Results

21.    L119, p should be italicized.

22.    L235, check sentence.

23.    L263, explain FDR

24.    L289-290, enzymes could be written with lowercase letters.

25.    L308-311, enzymes could be written with lowercase letters.

Discussion

26.    L329, write names of authors after by (“…. by X and Y [20-22])

27.    L331, by again.

28.    L343, change Tomato to tomato

29.    L343, change Plants to plants

30.    L372, Trichoderma should be italicized.

31.    L399-400, enzymes could be written with lowercase letters.

32.    L408, gene symbols should be written as italic.

33.    Author names of diseases should be written in the first mentioned place.

Conclusion

34.    L672-674, The last sentence could be given as a conclusion sentence in ABSTRACT.

Table S

35.    Change Tables to table.

References should be re-handled according to journal style.

Author Response

Response to Reviewer 1 Comments

Comments: Dear author(s),

The following points should clearly be corrected and explained for readers in the manuscript

 Response: I’m thankful to the reviewer. The point-by-point responses to each comment are listed below.

Comment 1 for Abstract

  1. L17, write a start sentence about importance of topic/study.
  2. L23, remove -
  3. L31, a conclusion sentence should be written for readers.

Response: I’m thankful to the reviewer. The abstract has been rewritten completely with taking your comments into consideration.

Comment 2 for key wards

  1. Scientific name of eggplant should be written

Response: Thanks for your carefulness. The scientific name of eggplant has been listed in Keywords as suggested.

Comment 3 for Introduction

  1. L35, author name of Verticillium dahliae should be written in the first mentioned place.

Response: Thanks for your carefulness. The modification has been made as you suggested

  1. L37, scientific name of eggplant and its author name should be written in the first mentioned place.

Response: Thanks for your carefulness. The modification has been made as you suggested

  1. L38, remove “… in Zhejiang province” because this disease is widespread in the world.

Response: Thanks for your carefulness. The phrase “… in Zhejiang province” has been removed.

  1. L56, change Transcriptome to transcriptome

Response: Thanks for your carefulness. The word “Transcriptome” has been changed to transcriptome.

  1. L59, T. virens should be italicized.

Response: Thanks for your carefulness. The “T. virens” has been italicized

  1. L60, author name of Rhizoctonia solani should be written in the first mentioned place.

Response: Thanks for your carefulness. The modification has been made as you suggested

  1. L63, author name of Pythium ultimum should be written in the first mentioned place.

Response: Thanks for your carefulness. The modification has been made as you suggested

  1. L65, Rhizopus oryzae should be italicized.

Response: Thanks for your carefulness. The “Rhizopus oryzae” has been italicized.

  1. L65, author name of Rhizopus oryzae should be written in the first mentioned place.

Response: Thanks for your carefulness. The modification has been made as you suggested

  1. L67, author name of Sclerotinia sclerotiorum should be written in the first mentioned place.

Response: Thanks for your carefulness. The modification has been made as you suggested

  1. L67, author name of Botrytis cinerea should be written in the first mentioned place.

Response: Thanks for your carefulness. The modification has been made as you suggested

  1. L68, a blank after dot.

Response: Thanks for your carefulness. The blank after the dot has been listed.

  1. L69, author name of T. harzianum should be written in the first mentioned place.

Response: Thanks for your carefulness. The modification has been made as you suggested

  1. L70-71, enzymes are not special name and they should be written with lower case letter.

Response: Thanks for your carefulness. An enzymes name has been written with lowercase letters.

  1. L75, change Proteins to proteins.

Response: Thanks for your carefulness. “Proteins” to “proteins” has been changed as suggested

  1. L78, Verticillium could be italicized.

Response: Thanks for your carefulness. The “Verticillium” has been italicized.

Comment 4 for results

  1. L119, p should be italicized.

Response: Thanks for your carefulness. The “p” has been italicized.

  1. L235, check sentence.

Response: Thanks for your carefulness. A sentence has been improved.

  1. L263, explain FDR.

Response: Thanks for your carefulness. The FDR has been explained.

  1. L289-290, enzymes could be written with lowercase letters.

Response: Thanks for your carefulness. Thanks for your carefulness. The enzyme names have been written in lowercase letters.

  1. L308-311, enzymes could be written with lowercase letters.

Response: Thanks for your carefulness. Thanks for your carefulness. The enzyme names have been written in lowercase letters.

Comment 5 for Discussion

  1. L329, write names of authors after by (“…. by X and Y [20-22])

Response: Thanks for your carefulness. The modification has been made as you suggested

  1. L331, by again.

Response: Thanks for your carefulness. The modification has been made as you suggested

  1. L343, change Tomato to tomato.

Response: Thanks for your carefulness. The “Tomato to tomato” has been changed.

  1. L343, change Plants to plants.

Response: Thanks for your carefulness. The “Plants to plants” has been changed

  1. L372, Trichoderma should be italicized.

Response: Thanks for your carefulness. The “Trichoderma” has been italicized.

  1. L399-400, enzymes could be written with lowercase letters.

Response: Thanks for your carefulness. Thanks for your carefulness. The enzyme names have been written in lowercase letters.

  1. L408, gene symbols should be written as italic.

Response: Thanks for your carefulness. The gene symbols have been italicized.

  1. Author names of diseases should be written in the first mentioned place.

Response: Thanks for your carefulness. But you did not specify the lines where the names of the diseases are located

Comment 6 for Conclusion

  1. L672-674, The last sentence could be given as a conclusion sentence in ABSTRACT.

 Response: Thanks for your carefulness. The last sentence of the conclusion has been listed in the abstract as you requested.

Comment 7 for Table S

  1. Change Tables to table.

Response: Thanks for your carefulness. The word "Tables" has been changed to "table" as you requested

Comment 8 for References

  1. References should be re-handled according to journal style.

Response: Thank you for drawing our attention. The references have been handled according to the Plants Journal style.

Reviewer 2 Report

Comments and Suggestions for Authors

The manuscrip presents the study of T. virens HZA14 potential to control Verticillium wilt disease by using diverse and comprehensive analyses starting with classical morphological methods, followed by pathogenicity tests in culture media and in planta, and supplemented with a series of molecular analyses.

The results are described clearly and in detail.

The discussion makes linkage between results that were obtained in the study and also with the data publish by other authors.

Materials and Methods are presented in detail.

The figures and tables are informative and well laid out.

References are up to date.

There are some comments and suggestions to the authors that are listed below. Several technical inaccuracies are pointed too.

Abstract

The abstract needs some improvement, as it does not clearly present the obtained results and their significance.

Page 1, rows 20-23: It is better to say “decreased disease severity up to 2.77%” instead of “decreased disease severity for 2.77%” or even emphasize that control efficacy is 96.59%.

The plant growth promotion effect of HZA14 has to be mentioned too.

Results

Page 2, rows 90-92

To which class belong HZA9 according to the results of the dual culture test?

Page 2, rows146-149

The measurement unit is missing after the values of disease severity and control efficacy.

Page 9, section 2.10

A lot of sentences contain “in the (9 d vs 3 d) included…” - brackets are redundant. Variants of the days have to be presented in the same manner as in the section 2.11, without brackets.

Discussion

Page 12, row 329: Instead of “results reported by [20-22]”, it is better to use “results reported by other authors [20-22]”. Similar on row 33o: “…the results studied by Jabnoun-Khiareddine [23]”.

Page 12, row 336: “….due to their having mechanisms” – not clear

Page 12, row 343: “tomato plants” instead of “Tomato plants”; “olive plants” instead of “olive Plants”

Page 12, row 344: “Many of the evaluated Trichoderma strains…..” instead of “Many of the Trichoderma strains evaluated….”.

Page 12, row 350: “….Fe3+ ions to Fe2+….” instead of “…Fe3+ ions to Fe2+…”

Page 12, rows 357-358: “…in agreement with results of our study …” instead of “…in agreement with our study results…”.

Page 12, row 372: Trichoderma – Italic

Page 13, row 393: “The diagrams on Figure 10 showed…” instead of “The diagrams in 10 showed…..”.

Conclusions

Page 19, row 666: “…structures of the fungal pathogen V. dahlia.” instead of “…structures of this fungal pathogen V. dahlia.”

Page 19, rows 669-672 – the sentence is too long and not clear enough

Comments on the Quality of English Language

Minor editing of English language required.

Author Response

Response to Reviewer 2 Comments

Comment: The manuscrip presents the study of T. virens HZA14 potential to control Verticillium wilt disease by using diverse and comprehensive analyses starting with classical morphological methods, followed by pathogenicity tests in culture media and in planta, and supplemented with a series of molecular analyses.

- The results are described clearly and in detail.

- The discussion makes linkage between results that were obtained in the study and also with the data publish by other authors.

- Materials and Methods are presented in detail.

- The figures and tables are informative and well laid out.

- References are up to date.

There are some comments and suggestions to the authors that are listed below. Several technical inaccuracies are pointed too.

Response: I’m thankful to reviewer. The point-by-point response to each comment is listed below.

Comment 1 for Abstract

1-The abstract needs some improvement, as it does not clearly present the obtained results and their significance.

2- Page 1, rows 20-23: It is better to say “decreased disease severity up to 2.77%” instead of “decreased disease severity for 2.77%” or even emphasize that control efficacy is 96.59%.

3- The plant growth promotion effect of HZA14 has to be mentioned too.

Response: I’m thankful to the reviewer. The abstract has been rewritten completely with taking your comments into consideration.

Comment 2 for Results

4- Page 2, rows 90-92: To which class belong HZA9 according to the results of the dual culture test?

Response: Thanks for your carefulness. Isolate HZA14 belonged to class 1. as mentioned in paragraph "2.1. Dual culture assay" (yellow colored).

5- Page 2, rows146-149: The measurement unit is missing after the values of disease severity and control efficacy.

Response: Thanks for your carefulness. The measurement units have been listed after the values of disease severity and control efficacy.

6-Page 9, section 2.10: A lot of sentences contain “in the (9 d vs 3 d) included…” - brackets are redundant. Variants of the days have to be presented in the same manner as in the section 2.11, without brackets.

Response: Thanks for your carefulness. The brackets in section 2.10 have been removed as you suggested.

Comment 3 for Discussion

7- Page 12, row 329: Instead of “results reported by [20-22]”, it is better to use “results reported by other authors [20-22]”. Similar on row 33o: “…the results studied by Jabnoun-Khiareddine [23]”.

Response: Thanks for your carefulness. The modification has been made as you suggested

8- Page 12, row 336: “….due to their having mechanisms” – not clear.

Response: Thank you for drawing our attention. The sentence has been improved.

9- Page 12, row 343: “tomato plants” instead of “Tomato plants”; “olive plants” instead of “olive Plants”.

Response: Thank you for drawing our attention. The words have been corrected as you suggested.

10- Page 12, row 344: “Many of the evaluated Trichoderma strains…..” instead of “Many of the Trichoderma strains evaluated….”.

Response: Thank you for drawing our attention. The phrase has been replaced as you suggested.

11- Page 12, row 350: “….Fe3+ ions to Fe2+….” instead of “…Fe3+ ions to Fe2+…”

Response: Thank you for drawing our attention. The Fe3+ ions to Fe2+ in row 350 has been replaced with the Fe3+ ions to Fe2+ as you suggested.

12- Page 12, rows 357-358: “…in agreement with results of our study …” instead of “…in agreement with our study results…”.

Response: Thank you for drawing our attention. The phrase has been replaced as you suggested.

13- Page 12, row 372: Trichoderma – Italic.

Response: Thanks for your carefulness. The "Trichoderma" have been italicized.

14- Page 13, row 393: “The diagrams on Figure 10 showed…” instead of “The diagrams in 10 showed…..”.

Response: Thank you for drawing our attention. The phrase has been corrected as you suggested.

Comment 4 for Conclusions

15- Page 19, row 666: “…structures of the fungal pathogen V. dahlia.” instead of “…structures of this fungal pathogen V. dahlia.”

Response: Thank you for drawing our attention. The phrase has been corrected as you suggested.

16- Page 19, rows 669-672 – the sentence is too long and not clear enough.

Response: Thanks for your carefulness. The sentence has been improved.

Comment 5: Comments on the Quality of English Language

17- Minor editing of English language required.

Response: Thanks for the valuable suggestion. The manuscript has been carefully revised by a native English Scientist and needful English, grammar, structural, and typographical mistakes have been removed as required by the reviewer 2.

Reviewer 3 Report

Comments and Suggestions for Authors

Dear Authors and Editor,

The English language needs to be thoroughly revised before submitting the paper for review. There are numerous grammatical errors in the abstract and first paragraph meaning I have not been able to review this paper.

Best wishes 

AM

Comments on the Quality of English Language

Dear Authors and Editor,

The English language needs to be thoroughly revised before submitting the paper for review. There are numerous grammatical errors in the abstract and first paragraph meaning I have not been able to review this paper.

Best wishes 

AM 

Author Response

Comment: Dear Authors and Editor,

The English language needs to be thoroughly revised before submitting the paper for review. There are numerous grammatical errors in the abstract and first paragraph meaning I have not been able to review this paper.

Response: Thanks for the valuable suggestion. The manuscript has been carefully revised by a native English Scientist and needful English, grammar, structural, and typographical mistakes have been removed as required by the reviewer 3.

Reviewer 4 Report

Comments and Suggestions for Authors

This manuscript entitled “ Potential of Trichoderma virens HZA14 in controlling Verticillium wilt disease of eggplant and analysis of its genes responsible for microsclerotial degradation” is of value to our knowledge on using Trichoderma spp as biological control against soil borne diseases. However, I have the following comments  on it.

Are the authors sure that there is no registered fungicide for wilt diseases?

The introduction reads as discussion. The authors should concentrate on the hypothesis of their study as well as the background of the studied topics (crop and pathogen). Discuss the previous results in their discussion and not in the introduction.

The authors has to explicitly explain why each tests / analysis in the materials and methods was made? Audience has to know the objective/s of those tests/analysis.

Authors advised to better compare their results with published data on Trichoderma from other countries ex Germany, New Zealand etc...also mention about the available Trichoderma products in the market!

Scientific writing should be revisited. Scientific names in the caption should be written in full. 

Comments on the Quality of English Language

I recommend authors improve the manuscript's English by a native English speaker.

Author Response

Response to Reviewer Comment

Comments: This manuscript entitled “Potential of Trichoderma virens HZA14 in controlling Verticillium wilt disease of eggplant and analysis of its genes responsible for microsclerotial degradation” is of value to our knowledge on using Trichoderma spp. as biological control against soil-borne diseases.

Response: I’m thankful to the reviewer for acknowledging my manuscript and suggesting the changes for further improvement.

Comment 1: However, I have the following comments on it.

Response: I’m thankful to reviewer. The point-by-point response to each comment is listed below.

Comment 2: Are the authors sure that there is no registered fungicide for wilt diseases?

Response: Thanks for your carefulness. Thanks for your carefulness. Modification and improvement of this sentence has been made.

Comment 3: The introduction reads as discussion. The authors should concentrate on the hypothesis of their study as well as the background of the studied topics (crop and pathogen). Discuss the previous results in their discussion and not in the introduction.

Response: Thanks for your carefulness. Modification has been made as suggested.

Comment 4: The author has to explicitly explain why each tests / analysis in the materials and methods was made? Audience has to know the objective/s of those tests/analysis.

Response: Thanks for your carefulness.  The cause made of each test/analysis in the materials and methods has been explicitly explained as suggested.

Comment 5: Authors advised to better compare their results with published data on Trichoderma from other countries ex Germany, New Zealand etc...

Response: Thanks for your carefulness. Improvement of discussion has been made.

Comment 6: Also mention about the available Trichoderma products in the market!

Response: Thanks for your carefulness. Some of the available Trichoderma products in the market have been listed.

Comment 7: Scientific writing should be revisited. Scientific names in the caption should be written in full.

Response: I’m thankful to the reviewer. The species names have been checked and corrected in the whole review manuscript.

Comment 8: I recommend authors improve the manuscript's English by a native English speaker.

Response: Thanks for the valuable suggestion. The manuscript has been carefully revised by a native English Scientist and needful English, grammar, structural, and typographical mistakes have been removed as required by the reviewer 4.

Round 2

Reviewer 4 Report

Comments and Suggestions for Authors

The authors have made substantial improvement on the manuscript and i recommend accept it for publication in Plants.